# Process Mining and Conformance Checking of Long Running Processes in the Context of Melanoma Surveillance [note 1]

**DOI:** 10.3390/ijerph15122809

**Published:** 2018-12-10

**Authors:** Christoph Rinner, Emmanuel Helm, Reinhold Dunkl, Harald Kittler, Stefanie Rinderle-Ma

**Affiliations:** 1Center for Medical Statistics, Informatics, and Intelligent Systems (CeMSIIS), Medical University of Vienna, Spitalgasse 23, Vienna 1010, Austria; 2Research Department of Advanced Information Systems and Technology, University of Applied Sciences Upper Austria, Softwarepark 13, Hagenberg 4232, Austria; emmanuel.helm@fh-hagenberg.at; 3Faculty of Computer Science, University of Vienna, Währinger Strasse 29, Vienna 1010, Austria; reinhold.dunkl@gmail.com (R.D.); stefanie.rinderle-ma@univie.ac.at (S.R.-M.); 4Department of Dermatology, Medical University of Vienna, Währinger Gürtel 18-20, Vienna 1010, Austria; harald.kittler@meduniwien.ac.at

**Keywords:** health care processes, process mining, electronic health records, medical guidelines

## Abstract

Background: Process mining is a relatively new discipline that helps to discover and analyze actual process executions based on log data. In this paper we apply conformance checking techniques to the process of surveillance of melanoma patients. This process consists of recurring events with time constraints between the events. Objectives: The goal of this work is to show how existing clinical data collected during melanoma surveillance can be prepared and pre-processed to be reused for process mining. Methods: We describe an approach based on time boxing to create process models from medical guidelines and the corresponding event logs from clinical data of patient visits. Results: Event logs were extracted for 1023 patients starting melanoma surveillance at the Department of Dermatology at the Medical University of Vienna between January 2010 and June 2017. Conformance checking techniques available in the ProM framework and explorative applied process mining techniques were applied. Conclusions: The presented time boxing enables the direct use of existing process mining frameworks like ProM to perform process-oriented analysis also with respect to time constraints between events.

## 1. Introduction

### 1.1. Motivation

The availability of electronic data in the early 1980s changed the business models of many companies, as well as the medical domain. The proliferation of electronic health data from a variety of sources including medical treatment records, administrative health data, and public health information systems supports the integration of all available evidence to feed back into medical research and practice. These health data collected during routine care can be used for secondary purposes such as identifying trends, predicting outcomes, influencing patient care, drug development, and therapy choices [1]. By analyzing these data it is possible to gain new insights which can be used for the optimization of health care processes, i.e., the insights might enable the transformation of health care processes instead of simply monitoring them.

Skin malignancies are recognized as a major and global health problem. Accounting for about 5% of all skin cancer cases, melanoma is the most dangerous form of skin malignancy and causes about 90% of skin cancer mortalities [2]. Incidence rates in many European countries are actually ranging between 12 and 15 cases per 100,000 inhabitants. Currently, the increasing rates are levelling off in some countries. In contrast, however, for distinct subpopulations, such as elderly men, the rates are still increasing [3]. Early detection of melanoma is of utmost importance and is leading to a favourable prognosis. Since melanoma may appear years after the excision of the primary tumor, patients with melanoma are monitored closely, usually following a predefined protocol, to allow timely detection of recurrent disease [4].

### 1.2. Problem Statement

In order to improve the surveillance of patients with melanoma, traditional studies depended on manual data acquisition. The goal of this work is to show how existing data from routine care can be combined from different sources and reused for process mining to automatically detect processes [5] and compare them to medical guidelines using conformance checking [6,7].

Process mining is a relatively new discipline that helps to discover and analyze actual process executions based on log data. Log data stores events that are produced during process execution. For example, one case of a patient in a hospital creates a trace (i.e., a sequence of events) comprising records of the execution of different process activities, e.g., “excision”. Process mining techniques are particularly promising to be applied in the healthcare domain facing challenges to (a) learn the process of interest; (b) understand deviations, (c) analyze bottlenecks, and (d) monitor organizational behaviour [8].

### 1.3. Research Questions

This paper focuses on challenges (a) and (b) and aims at learning about the applicability and results of process mining and conformance checking for the treatment and surveillance of melanomas. Previous studies have indicated the potential of process mining in the area of the surveillance of patients with melanoma [9,10]. In these two studies, however, only a maximum of 10 process instances were analyzed. Moreover, several data challenges were pointed out, specifically that the time granularity of the logged data was too coarse [9]. This problem is also referred to by [8] as imprecise data. Hence, this paper addresses the following research questions:
How can existing clinical data be reused for the application of process mining?How can data of recurring events with time constraints that span a long period of time be prepared to apply process mining?How can we apply process mining to check guideline compliance?What can we learn from process mining in the context of surveillance of melanoma patients?

### 1.4. Contribution

This paper provides conceptual extensions towards data preparation for imprecise log data. In particular, we describe a method for data preparation using a specific labelling convention to model the time aspects used in medical guidelines (e.g., check up in 6 months). The methods were tested using follow-up guidelines for melanoma patients [2] and anonymous patient data from the Department of Dermatology at the Medical University Vienna (DDMUV).

In Section 2 we give a brief overview of the process mining discipline and the positioning of this paper in the field of process mining in healthcare. In Section 3 we describe the data preparation and time boxing steps as well as the conformance checking applied in this paper. In Section 4 the study population at the DDMUV is described, time boxing and conformance checking are applied to melanoma surveillance data and the medical implications are analyzed. In Section 5 the generalized applicability of our methods are discussed.

## 2. Related Work

### 2.1. Process-Oriented Analysis

Process mining [5] offers techniques for different analysis tasks such as *discovery* of process models, *conformance checking* between process execution logs and process models or guidelines, and *enhancement* of existing models using information about the recorded reality in process execution logs (i.e., event logs). Event logs store events that are produced by the execution of the process tasks during runtime. Existing standardized formats for event logs are MXML and XES [11]. In this article we mainly focus on the task of *conformance checking*.

There are different techniques and algorithms to measure conformance (e.g., [7,12]). The degree of conformance is measured in four orthogonal dimensions. (1) Fitness is the most agreed upon measure to determine if the model reflects the recorded behaviour in the event log. Approaches to measure fitness are listed and evaluated by [13]. (2) Precision aims to identify overly general models and penalize unwanted behaviour. A recent study, however, shows that for all existing precision metrics there are cases where they do not provide the desired properties to reliably recognize under-fitting [14]. (3) Generalization measures the degree of overfitting, i.e., if the model only allows for what has actually been observed. (4) Structural appropriateness, derived from the simplicity quality dimension described in [5], aims to find an easy to understand and not overly complex model.

### 2.2. Positioning in the Field

Rojas et al. provide a recent survey on existing literature and case studies on process mining in healthcare, collecting 74 publications in this area [15]. Following their terminology, our paper analyzes *organizational* processes based on data from a *clinical support system* for *oncology* in *Austria*. It poses a *specific question* (i.e., guideline compliance) and utilizes the *conformance* perspective. This paper presents a *semi-automated* implementation strategy, providing a novel data preparation approach facilitating the use of the tool ProM. The analysis follows the basic approach, using the ProM Multi-perspective Process Explorer (MPE) plugin (Process Mining Group, Eindhoven Technical University, Eindhoven, The Netherlands) [6].

## 3. Methods

The process of surveillance of melanoma patients starts with the detection and the excision of the primary tumor (i.e., the baseline visit). Melanoma patients are staged according to the American Joint Committee on Cancer (AJCC) staging system (i.e., stage I to IV). After excision of the primary tumor patients start a 10 year surveillance period. Depending on AJCC staging, follow-up visits have different surveillance intervals and include different types of examinations (i.e., clinical examination, analyzing tumor markers, lymph node sonography, computed tomography of the abdomen, PET-CT etc.). In AJCC stage I, for example, the interval of the follow-up visits is 6 months in the first 5 years and one year between the 5th and the 10th year. In the other AJCC stages, intervals of 3 months in the first five years and 6 months between the 5th and the 10th year are scheduled. The higher the AJCC stage, the more often examinations are performed as part of a follow-up program. During surveillance, the AJCC stages are re-evaluated and patients can be assigned a higher AJCC stage and start the corresponding follow-up surveillance from the beginning. In this paper we refer to this upgrading as stage change. Since the observation period of this study only covers 7 years (January 2010 to June 2017), patients are still compliant to the guideline if they have not missed the next-to-last or last follow-up visit before the end of the study (i.e., June 2017). Patients are considered lost to follow-up when the surveillance is terminated prematurely at the DDMUV (e.g., patient changed clinic). The events occurring during melanoma surveillance are depicted in Figure 1.

We took the clinical data needed to perform the process mining from a local melanoma registry stored in the Research, Documentation and Analysis (RDA) Platform of the Medical University of Vienna [16]. Additional information about the transfer of patients between different clinics within the hospital, laboratory results, as well as treatment information is obtained from the local Hospital Information System (HIS). Since the melanoma registry is maintained manually the information from the HIS is used to detect additional follow-up visits re-using information from routine care. The event logs used in this study are created using the JAVA programming language, a JDBC driver to access the data in the Oracle Database, and the OpenXES library to create event logs in the MXML format. Conformance checking was performed in the ProM framework. The study was approved by the ethics committee of the Medical University of Vienna (EK Nr.: 1297/2014).

The initial explorative analysis of the process model and evaluation purposes were conducted with the commercial process mining tool Disco [17]. The extended fuzzy miner of Disco allows for fast prototyping and quick results.

### 3.1. Data Preparation and Time Boxing

According to the guideline, depending on the AJCC stage of the patient, the follow-up treatment takes place at certain time intervals (i.e., every three, six, or twelve months) in a repeated fashion for ten years. The existing process execution logs record the same event for each occurrence (e.g., *follow-up visit* for each follow-up visit), so the process mining algorithms are not able to distinguish between these events depending on the fixed time period specified in the guideline (e.g., second follow-up visit after one year).

Using the simplified process model depicted in Figure 1, it is not possible to distinguish the different follow-up visits automatically and as a consequence conformance to the guideline cannot be checked using current process mining algorithms. To overcome this problem we propose a labelling convention based on *time boxing* for recurring events commonly described in medical guidelines. During the time boxing, each activity (e.g., each follow-up visit) is allocated (i.e., aligned) to a predefined fixed time period it matches in, called a time box. Each time box corresponds to an event in the medical guideline and the events in each time box are labelled according to the label of the time box (e.g., *I_F_01_1Q* corresponds to an AJCC stage I follow-up visit in the first quarter of the first year). In Figure 2 the process model with applied time boxing corresponding to the melanoma surveillance guideline used at the DDMUV is shown.

The event log follows the same labelling convention as the process model. To generate the event log, all follow-up visits are assigned to the corresponding (i.e., temporally closest) time boxes. All follow-up visits in one time box are merged and represented as one. In the guidelines, a single visit in a specific time period is recommended. In order to analyze over-compliance (i.e., more than one visit of a patient during a specific time period instead of a single visit as stated in the guideline), multiple events could be assigned to the same time box (without merging) and the resulting self-loops considered during the analysis.

### 3.2. Conformance Checking with ProM

The conformance checking was done using the process mining framework ProM in version 6.6 and the respective plug-in *Multi-perspective Process Explorer* (MPE) [6]. It allows for fitness and precision calculation and provides different views on the data, including (1) a model view, depicting the base model petri net, (2) a trace view, making it possible to investigate individual traces, and (3) a chart view, showing the distribution of attribute values in the log for certain parts of the model.

The MPE is an advanced tool that integrates state of the art algorithms described in [18] (fitness) and [19] (precision), that are also able to integrate different perspectives i.e., data, resource, and time. The configuration for penalties on log and event moves was adapted to the specific use case. A valid configuration for the alignment parameters (penalties for moves on the log/model) had to be identified. Due to the pre-processing there are no wrong events (events present in the log but not in the model) save for the LTFU (i.e., Lost to follow-up) event, so the alignment algorithm must always identify the missing events (events in the model that are missing in the log).

For the alignment, the default costs of the MPE for missing events in the log (value: 2) and missing activities in the model (value: 3) leads to undesired behaviour. The alignment algorithm identifies follow-up visits after a long period of time as wrong events. When the penalty for a sequence of missing events exceeds the penalty for a wrong event, the alignment algorithm will declare the current event wrong. In order to ensure a correct alignment, the maximum number of skipped follow-up visits in all traces was identified and the penalties adopted respectively. Since the maximum number of consecutive skipped events for one trace is 19 in our data, we chose a penalty of 1 for missing events and 20 for wrong events. For the LTFU event we reduce the wrong event penalty to 0, thus only penalizing the missing IN_FUP event at the end and not overvalue the outcome indicator for the fitness calculation.

### 3.3. Statistical Methods

To evaluate the medical implications R (v3.4.4) and the Kaplan–Meier estimator for survival analysis (survival v2.41, Hmisc v4.1, Survminer v0.4) were used. For the survival analysis the fitness is split into three equal-sized groups.

## 4. Results

The DDMUV is a tertiary referral centre that offers a long-term surveillance program for melanoma patients based on the European guideline on melanoma treatment [2]. An example for the follow-up sub process in the European guideline on melanoma treatment modelled in the Business Process Modeling and Notation (BPMN) can be found in [9]. The melanoma registry at the DDMUV contains data of baseline and follow-up visits of melanoma patients. Excisions are documented way back to the early 1990s, a continuous documentation of the follow-up visits started 2010. In 2017, the melanoma registry covered about 2200 patients. In this study we included all 1023 patients (43% females, mean age 59±17.5 years) with baseline visit (i.e., excisions) after January 2010 and at least one follow-up visit since patients without a single follow-up visit only had the excision at the DDMUV and no data is available in the melanoma registry. Besides the demographic data, different characteristics of the identified melanoma are documented. For the baseline visit this includes among others, (1) melanoma subtype (superficial spreading melanoma, nodular melanoma, lentigo maligna melanoma, acral lentiginous melanoma and others), (2) anatomic site (i.e., abdomen, hand, foot, head, etc.), (3) depth of invasion, (4) date of surgery for the primary tumor, and (5) staging information. More than one primary tumor can be documented. Only melanoma staging is used for conformance checking. We extracted five different event logs from our real world data, one including all patients (i.e., I-IV), and four for each AJCC stage separately (i.e., I, II, III, IV) based on the highest AJCC stage of the patient. If a patient initially started with AJCC stage I and then moved to AJCC stage II the patient is represented in the AJCC II log file. Table 1 lists the number of patients and number of mean events per case in each log.

The number of patients per AJCC stage decreases with higher AJCC stage, which corresponds to the fact that most melanomas in Austria are diagnosed in early stages [20]. Most patients (n=401) were in AJCC stage I. This group also had the highest number of patients lost to follow-up (n=313, 78%). The ratio of patients IN_FUP (i.e., in follow-up) was the highest in AJCC stage IV with 58% (n=100). There is no difference between proportion of individuals lost to follow-up (LTFU) between men and women. Men were generally older than women and there was no significant difference between the LTFU and IN_FUP in respect to the age. Patients in lower AJCC stages were generally younger (I: Mean age 57±17 years; II: Mean age 59±18 years; III: Mean age 60±18 years; IV: Mean age 63±16 years).

### 4.1. Conformance Checking of Melanoma Surveillance

The results of the conformance checking in the form of fitness and precision indicators can be seen in Table 2. Our measurements show that the guideline models have an overall comparable and good fitness value, i.e., the model generally explains the behaviour seen in the log. This originates from three facts. (1) The relabelling and clustering of activities was done based on the terminology that was also used for the guideline model. (2) The time boxing method presented in Section 3.1 leads to an ordered sequence of events, where loops and duplicates cannot occur. (3) The only *wrong* events (i.e., events present only in the log, not in the model) are the LTFU events.

The precision of the model for stage I is 75.1% and declines to 63.1% for stage IV. The ratio between observed and possible behaviour indicates under-fitting for low values. The explanation for the generally lower precision values is that the guideline models include the whole time period of ten years of follow-up visits, while the event logs only cover a maximum of seven and a half years. Thus, modelled events like I_F_08_1Q (i.e., stage I, eighth year, first quarter) will never be reached during replay, leading to a lower precision. The explanation for the declining values of precision is that the guideline models for higher stages allow for all the lower stages’ events too, since a patient can start in stage I and be re-evaluated to stages II, III, or IV during their follow-up visits. The number of possible behaviour is thus higher while the number of actual patients in the stages is similar (II) or significantly lower (III and IV) than in stage I.

Figure 3 shows the most frequent trace recorded in the complete log. One hundred and forty-eight of the 1023 patients follow this trace where they (1) start with the excision (Start), (2) are staged in AJCC I (StageChange), (3) go to their first follow-up (I_F_00_3Q), and (4) are afterwards lost to follow-up (wrong event LTFU). The following missing event (IN_FUP) is in the guideline model but was not present for those traces in the log. Finally, the End event concludes the trace.

Figure 4 shows a patient that started in stage I and was re-evaluated to stage II and later to stages III and IV. All in all just 1 follow-up visit during stage II was missed and the fitness is very high. The trace spans over the whole observation period, with the start in 2010 and the last follow-up in late 2015. Thus, the patient was identified as in follow-up (IN_FUP).

In Figure 5 the patient classified in stage II skipped multiple follow-up visits and left the monitoring entirely after four years. The low fitness value correlates with the low guideline compliance.

### 4.2. Applied Process Mining

In addition to conformance checking we applied several techniques and tools to the data at hand in an explorative manner, to try and find interesting trends in the data and to validate our results with domain experts. Figure 6 shows the dotted chart analysis of the stage I event log. For stages II–IV the number of distinct event types becomes higher and additional stage changes speckle the diagram so the rainbow pattern becomes less noticeable.

The recording period of 7.5 years, from January 2010 to June 2017, can be seen on the y-axis. Observations made based on this dotted chart:
A clear ‘rainbow’ pattern is visible since the sequence of events recorded for stage I is linear and follow-up visits occur at generally regular intervals.The outcome indicator (IN_FUP/LTFU event) is not visible since the End event occurs at the same timestamp +20 h and the latter dot overlaps the former one.The most frequent trace (i.e., ending the follow-up after the first visit in stage I—see Figure 3) can be observed due to the high number of End events that occur shortly after the Start and I_F_00_3Q events.

Figure 7 shows a part of the model depicting the flow of patients in stage I. It also includes the lost to follow-up step (LTFU) that marks an early dropout. Observations that can be made on this model section:
All 401 patients in stage I start with the ‘I_F_00_3Q’ event. One hundred and forty-eight patients drop out (i.e., LTFU) after this step. This corresponds to the 148 patients of the most frequent trace in Figure 3.The frequency of following events declines steadily. There are fewer patients in later steps than at the beginning. There are two reasons for that. (1) Not all patients started at the same time, thus not all patients can reach the 7th years’ follow-up visits in the fixed time interval. (2) After each step some of the patients drop out.The sequence of follow-up events is not linear but makes a ‘braided’ impression due to the skipping of single follow-up visits (see also Figure 5).

### 4.3. Medical Implications

In [21] the prognosis among patients with thin melanomas depending on the surveillance compliance was analyzed. Patients were considered to be compliant with the follow-up regimen if they had at least one annual follow-up examination and non-compliant if they had follow-up intervals of more than one year. They showed that compliant patients before the onset of recurrence had a significantly better prognosis than non-compliant patients.

When using our calculated fitness instead of the fixed time-intervals to evaluate the survival, the same effect can be observed in our data as seen in Figure 8. We sampled all 246 patients that stayed in follow-up for more than two years (a subset of Table 1, Total, IN_FUP, I-IV) based on their fitness value into three equal-sized groups and used the Kaplan–Meier estimator for survival analysis. The survival probability of patients with a high guideline compliance after five years is about 5% higher compared to the least compliant group. However, adding the patients that stayed for less than 2 years to the estimator, looking at all 358 patients in follow-up (Table 1, Total, IN_FUP, I-IV), showed a reversed effect. The main reason was that higher fitness is easier to achieve with a shorter stay and many with a short stay died early, e.g., after being staged in IV and the first follow-up visit.

## 5. Discussion and Lessons Learnt

### 5.1. Reuse of Clinical Data for Process Mining

We reused existing patient data available from a local electronic health record system in the context of melanoma surveillance. In combination with the local melanoma registry additional follow-up visits and laboratory data to the event log were identified. Creating the log file using a procedural programming approach allowed us to add pre-processing steps. For example, we tagged patients that successfully terminate the process (i.e., still in the surveillance program (IN_FUP)) during the creation of the log file based on the time they did not show up before the end (i.e., time boxes missed). Beside the melanoma registry, more than 70 other registries are documented in the RDA platform. In a current master thesis, a mapping of the melanoma registry data from the RDA data model [22] to the i2b2 star schema [23] and the OMOP common data model [24] is performed. By adapting our approach to these two widely used data models, a greater variety of data could be made available to process mining and conformance checking in particular.

### 5.2. Events with Time Constraints Spanning a Long Period of Time

To perform conformance checking, events in event logs are compared to activities in a process model. Our process model, the melanoma guidelines, covers a time period of 10 years, while the event log is only 7.5 years since the melanoma registry only covers 7.5 years. These long running processes are common in the medical domain and have to be considered when modelling the process model. Our approach of time boxing, by coding AJCC stage information and the time dimension into the event names, enabled us to use conformance checking based on imperative Petri net models. We were able to check if patients’ follow-up visits were conforming to the time frames given by medical guidelines. Our approach was solely used to check whether a patient attended a specific follow-up visit. Generally, this way of pushing additional information into the model can be widely used to answer diverse questions; the structure of the generated log files has to be adapted to the task at hand. The labelling convention using time information and the pushing additional data into the event names leads to bloated models and adds complexity to the model. In order to prevent uncontrolled growth of number of events, a higher granularity of time information was used. Depending on the AJCC stage the granularity is “three months”, “six months”, or “one year”. The model with time boxes only consists of linear paths. To ensure consistent order of events on the same day (i.e., the stage change is detected at the same day as the follow-up visit), we applied activity sequencing by applying a hard coded time of day for each type of event (Stage change always at 8 a.m., follow-up-visit at 10 a.m. etc.). By only coding the lost to follow-up into the event logs and not into the model, we could easily penalize lost to follow-up. Lost to follow-up is the only “wrong event”. The calculation of the precision does not yet take into account the period of time the patient has been under surveillance. In order to get more meaningful results the expected model could be adjusted to the period that each patient has been under surveillance.

### 5.3. Guideline Compliance Checking

In our approach, we pushed the time dimension into the process structure to be able to use the conformance checking capabilities of ProM on imperative models. However, there are two viable alternative approaches: (1) Using a data-aware alignment algorithm would allow to keep the time dimension hidden in the data, thus labelling the follow-up visits just *follow-up*, avoiding the initially confusing time boxing notation (e.g., I_F_00_3Q). However, we decided to use our labelling approach to make all (missing) steps easily visible in the model. (2) The current version of the ProM framework also includes a declarative mining module that derives sets of constraints in form of a declarative model from log files and offers also conformance checking [25,26]. This needs further investigation, especially in the preparation of a correct declarative constraint set based on the guideline, as well as an adapted real log to be replayed.

Artificial neural networks have been used to classify images of skin lesions for many years. Newer neural networks (i.e., after 2012) use convolutional neural networks which apply reusable filters that simplify the network connections and are hence more suitable for image classifications. In a recent retrospective study, we used this technique to detect skin cancer [27,28]. Robust artificial neural networks can help to detect skin cancer in an earlier stage and improve the outcome for the patients. Future neural networks could help to triage patients and support physicians with their guideline compliant treatment.

The compliance calculated and formalized using the fitness using MPE is very promising. Yet it has to be further analyzed under which circumstances it correlates to the outcome of the patients. Further, we plan to analyze how the compliance affects the tumor progression of the patients, i.e., if patients with a higher compliance are less likely to progress to a higher AJCC stage.

## Figures and Tables

**Figure 1 ijerph-15-02809-f001:**
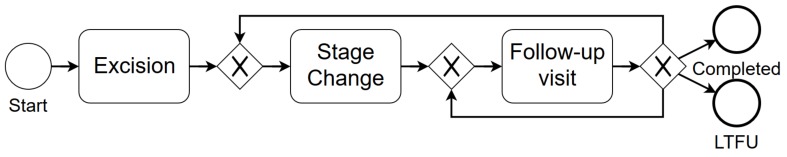
A Business Process Modeling and Notation (BPMN) representation of the process of melanoma surveillance. The start event of the surveillance is the excision of the melanoma followed by the American Joint Committee on Cancer (AJCC) stage classification and the follow-up visits. Depending on the AJCC stage, the number of follow-up visits can vary and the AJCC stage can be re-assessed after each follow-up visit. A patient can be lost to follow-up or complete the surveillance successfully.

**Figure 2 ijerph-15-02809-f002:**
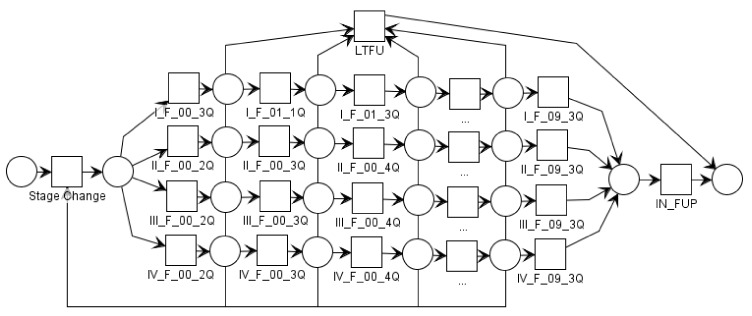
A simplified petri net model with applied time boxing corresponding to the guideline used at the Dermatology at the Medical University Vienna (DDMUV). For each AJCC stage (i.e., I, II, III, and IV) the follow-up visits after three, six, or twelve month (i.e., 2Q is 2nd quarter, 3Q is 3rd quarter, and 4Q is 4th quarter) for ten years are shown. After each follow-up visit a patient can proceed to any later follow-up visit, have a state change, be lost to follow-up (LTFU), or complete the surveillance (i.e., IN_FUP meaning *still in follow-up*).

**Figure 3 ijerph-15-02809-f003:**
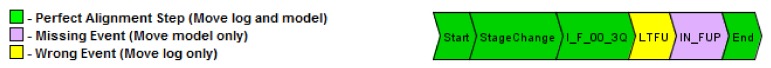
The most frequent trace in the complete log (I–IV) visualized via the Multi-perspective Process Explorer’s (MPE) trace view (fitness 98.8%).

**Figure 4 ijerph-15-02809-f004:**
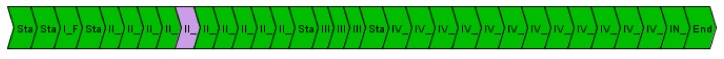
A trace comprising all four stages and only one missing follow-up visit (99.8%).

**Figure 5 ijerph-15-02809-f005:**
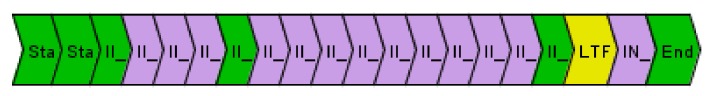
A trace with multiple skipped events and thus relatively low fitness (89.6%).

**Figure 6 ijerph-15-02809-f006:**
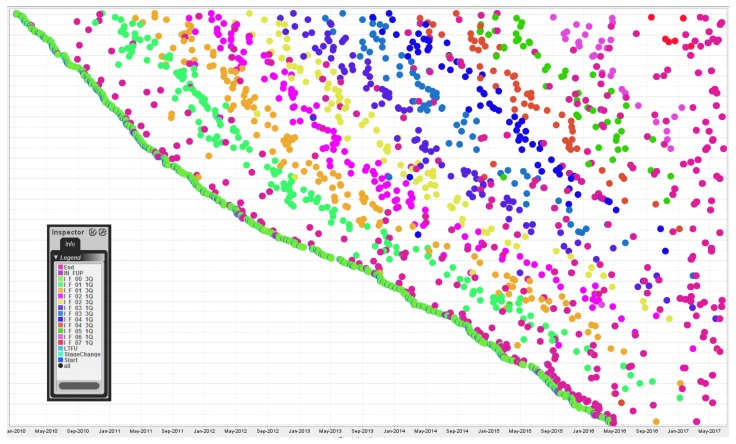
Dotted Chart Analysis of the stage I event log. The x-axis shows the timestamps of the events. The y-axis lists all cases sorted by the timestamps of their start event, descending. The dots represent the events, color-coded based on event label according to the legend on the left.

**Figure 7 ijerph-15-02809-f007:**
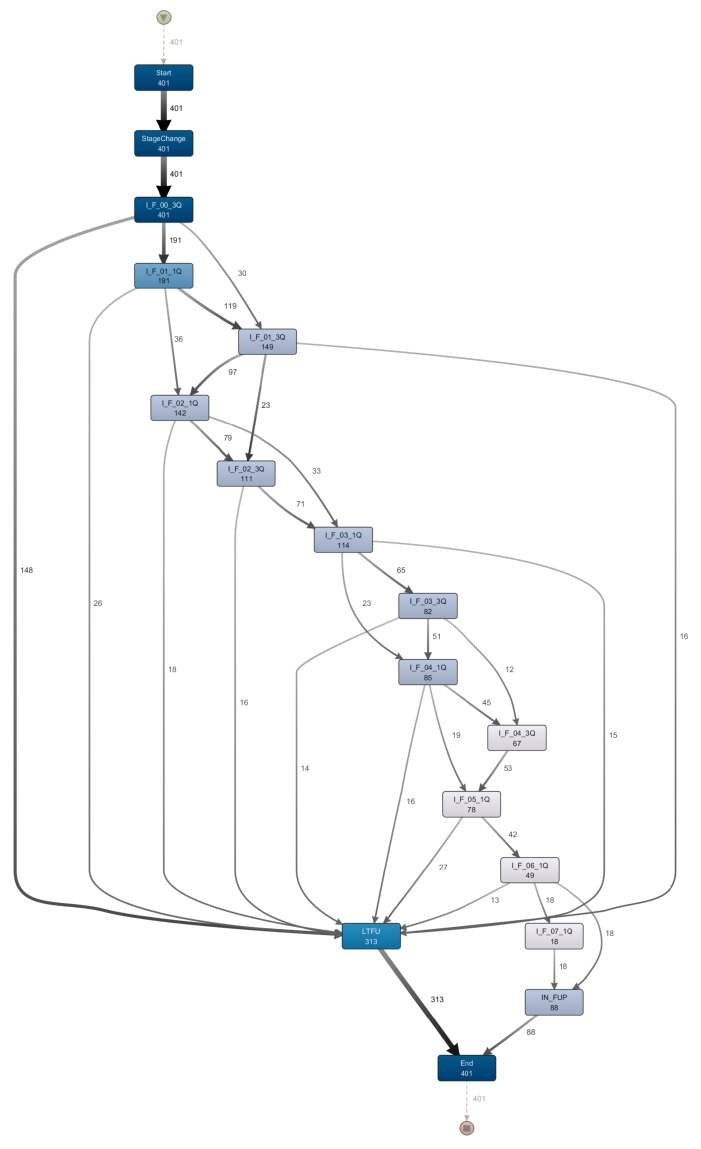
A section of a model created using the process mining tool Disco (Fluxicon BV, Bomanshof, Eindhoven, The Netherlands). The model is based on the stage I log. The tool was set to show all activities (100%) and the most frequent pathways (30%). The colour of the events (dark—more, light—fewer) and the thickness of the pathways (thicker—more) represent the frequency in the log.

**Figure 8 ijerph-15-02809-f008:**
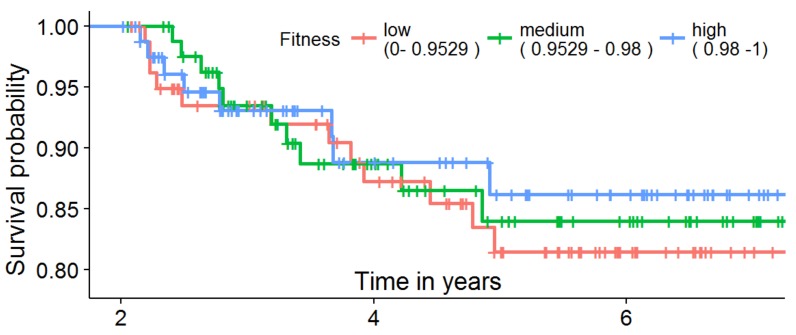
Survival analysis for all 246 patients that stayed in follow-up for more than two years, sampled into three equal-sized groups depending on their fitness.

**Table 1 ijerph-15-02809-t001:** Number of patients and mean number of events per case in log files split by maximal American Joint Committee on Cancer (AJCC) stage and separated by LTFU (lost to follow-up) and IN_FUP (in follow-up). The distribution of the outcome indicators LTFU and IN_FUP show significant differences between the AJCC stages.

		Number of Patients					Number of Mean Events per Case				
		I–IV	I	II	III	IV	I–IV	I	II	III	IV
	LTFU	286	146	95	20	25	8.23	6.78	9.2	10.6	11.2
Female	IN_FUP	153	45	50	17	41	12.8	10.1	14.9	13.4	13
	Total	439	191	145	37	66	9.82	7.57	11.2	11.9	12.3
	LTFU	379	167	124	42	46	8.67	7.09	8.90	10.7	11.9
Male	IN_FUP	205	43	70	33	59	13.3	10.7	13.9	15.1	13.4
	Total	584	210	194	75	105	10.3	7.84	10.7	12.6	12.7
	LTFU	665	313	219	62	71	8.48	6.95	9.03	10.7	11.6
Total	IN_FUP	358	88	120	50	100	13.1	10.4	14.3	14.5	13.2
	Total	1023	401	339	112	171	10.1	7.71	10.9	12.4	12.6

**Table 2 ijerph-15-02809-t002:** Using the Multi-perspective Process Explorer (MPE), for each stage log and the combined log (I–IV) the average fitness and precision were calculated in regard to the respective guideline models.

AJCC Stage	No of Patients	Avg. Fitness % (Min–Max)	Precision % (#Observed / #Possible Behaviour)
I	401	98.6% (91–100%)	75.1%
II	339	98.0% (82–100%)	71.4%
III	112	98.2% (85–100%)	65.0%
IV	171	98.7% (88–100%)	63.1%
I–IV	1023	98.4% (53–100%)	87.0%

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
