# Peer review of "Process Mining and Conformance Checking of Long Running Processes in the Context of Melanoma Surveillance"

_ijerph, 2018, doi:10.3390/ijerph15122809_

Reviewer 1 Report

The article proposes a method to apply conformance checking to analyze the process of surveillance of patients with melanoma. An event log is generated based on clinical records, in which each event represents the patient's attendance to a follow-up control in a certain period (based on quarters) after a first melanoma excision. This is compared to a reference process model created from medical guidelines.

The idea of applying conformance checking for the analysis of the process of surveillance of patients with melanoma is very promising, in particular, the approach used for data pre-processing, and the idea of discretizing the time based on time boxing. The examples shown in Figures 4, 5 and 6, illustrate that the results are meaningful from a surveillance compliance point of view.

The document is in general well written.

I also reviewed the article also for PODS4H. I liked the improvements the authors made for this version of the article:

- Present models in standard notations: BPMN and Petri nets

- Include results using other process mining tools, such as Dotted Chart Analysis and DISCO

- The new subsection 5.2 “Events with time constraints spanning a long period of time” highlights some of the main contributions of the article for future research

- I am glad the authors have found under what conditions the overall fitness results is consistent with medical expectations, i.e., compliant patients should have a better prognosis

I believe that the expected model should be adjusted to the period that each patient has been under surveillance, so that the precision results be more meaningful for the analysis of patient behavior. At least, this should be considered as a future work.

More specific details

- Abstract: it can be extended to include the improvements the authors made for this version of the article.

- In subsection 1.3: it is mentioned that there are few applications of conformance checking in “this area”. Please be more specific and refer to “the surveillance of patients with melanoma”.

- In subsection 3.1: the last idea is not clear: "In order to analyze over-compliance, ..."; I think it would be useful to explain it better.

- line 330: there is a typo: “Department62of” => “Department of”.

- line 332: the acronym FUP is never used alone; it makes more sense to consider the acronym IN_FUP.

Author Response

The authors would like to thank the editor and the reviewer for their precious time and comments. We have carefully addressed all the comments. The corresponding changes and refinements made in the revised paper are summarized in our response below.

Reviewer 1:

Comments and Suggestions for Authors

The article proposes a method to apply conformance checking to analyze the process of surveillance of patients with melanoma. An event log is generated based on clinical records, in which each event represents the patient's attendance to a follow-up control in a certain period (based on quarters) after a first melanoma excision. This is compared to a reference process model created from medical guidelines.

The idea of applying conformance checking for the analysis of the process of surveillance of patients with melanoma is very promising, in particular, the approach used for data pre-processing, and the idea of discretizing the time based on time boxing. The examples shown in Figures 4, 5 and 6, illustrate that the results are meaningful from a surveillance compliance point of view.

The document is in general well written.

I also reviewed the article also for PODS4H. I liked the improvements the authors made for this version of the article:

- Present models in standard notations: BPMN and Petri nets

- Include results using other process mining tools, such as Dotted Chart Analysis and DISCO

- The new subsection 5.2 “Events with time constraints spanning a long period of time” highlights some of the main contributions of the article for future research

- I am glad the authors have found under what conditions the overall fitness results is consistent with medical expectations, i.e., compliant patients should have a better prognosis

→ We added the following to the discussion: “The calculation of the precision does not yet take into account the period of time the patient has been under surveillance. In order to get more meaningful results the expected model could be adjusted to the period that each patient has been under surveillance.”

More specific details

· Abstract: it can be extended to include the improvements the authors made for this version of the article.

→ We added the explorative applied process mining techniques to the abstract

· In subsection 1.3: it is mentioned that there are few applications of conformance checking in “this area”. Please be more specific and refer to “the surveillance of patients with melanoma”.

→ We added “the survveillance of patients with melanoma”.

· In subsection 3.1: the last idea is not clear: "In order to analyze over-compliance, ..."; I think it would be useful to explain it better.

→ We added an explanation to over-compliance (i.e. more than one visit of a patient during a specific time period instead of a single visit as stated in the guideline)  

· line 330: there is a typo: “Department62of” => “Department of”.

→ Typo was corrected

· line 332: the acronym FUP is never used alone; it makes more sense to consider the acronym IN_FUP.

→ We changed the abbreviation FUP to IN_FUP

Reviewer 2 Report

Process mining, an emerging data analytics method, has been used effectively in various healthcare contexts including oncology. In this study, it proposed the process model as ''time boxing" to describe the patient's tracking status, and then use the log data to perform process mining as well as conformance checking.  Some comments are as follows.

1. In the first paragraph in Results section, the patient's AJCC stage changed (i.e. I to II), and his or her log data will be represented in the new stage. However, the precision of the model declines from stage 1 to stage 4 (Table 2).  It indicated that the model might be not suitable if the patient occurs an AJCC stage change. Or the model requires to be revised to present a more precise representation of reality. Please explain more details in this section.

2. In section 'Medical implications', the authors state that: " We sampled all 246 patients.....more than two years....". And in the following sentences, the authors state that: "...adding the patients....less than 2 years to the estimator, looking at all 358....". It is unclear whether the study cohort consists of 1,023 subjects or ? subjects.

Author Response

The authors would like to thank the editor and the reviewer for their precious time and comments. We have carefully addressed the comments. The corresponding changes and refinements made in the revised paper are summarized in our response below.

Comments and Suggestions for Authors

Process mining, an emerging data analytics method, has been used effectively in various healthcare contexts including oncology. In this study, it proposed the process model as ''time boxing" to describe the patient's tracking status, and then use the log data to perform process mining as well as conformance checking. Some comments are as follows.

1. In the first paragraph in Results section, the patient's AJCC stage changed (i.e. I to II), and his or her log data will be represented in the new stage. However, the precision of the model declines from stage 1 to stage 4 (Table 2). It indicated that the model might be not suitable if the patient occurs an AJCC stage change. Or the model requires to be revised to present a more precise representation of reality. Please explain more details in this section.

→  In Section 4.1 we use one paragraph to explain the snapshot problem (i.e. 10 year guideline vs. 7.5 years of data). We added the following to the discussion as future work:  “The calculation of the precision does not yet take into account the period of time the patient has been under surveillance. In order to get more meaningful results the expected model could be adjusted to the period that each patient has been under surveillance.”

2. In section 'Medical implications', the authors state that: " We sampled all 246 patients.....more than two years....". And in the following sentences, the authors state that: "...adding the patients....less than 2 years to the estimator, looking at all 358....". It is unclear whether the study cohort consists of 1,023 subjects or ? subjects.

→ As stated in Table 1, 358 patients are IN_FUP (I-IV). Only 246 of these were IN_FUP more than two years. We added a some description and a reference to the table (i.e. subset of \ref{tab:stages}, Total, IN\_FUP, I-IV) to clarify this part.

Reviewer 3 Report

Rinner et al. Present a study of conformance checking in the follow-up of Melanoma using process mining techniques. The paper is well written and it is interesting. I enjoyed reading it and I am quite confident it can help epidemiologists, statisticians, doctors and engineers in producing advanced decision support systems.

I have only one major concern and it is about the formal structure as a scientific paper. The main shortcoming I have detected during the review is the thread between methods and results. On its current form, the manuscript presents and describes some results which were not presented in the objective nor the methodology, leading to a potential confusion and lack of reproducibility. Authors should complete the methods section to describe: 1) Precision and fitness measurements 2) penalties (weights) configuration (line 153) 3) define what a trace  is and what an event is 4) Define how you aim to analyse survival 5)Define why, how and for what did you used DISCO instead of ProM. Finally, discussion section should contain the authors viewpoint of the results and comparison with the related work and state of the art. Including in the discussion new information (Fig 8) is misleading. Please move this chart and its description to results and only interpret it in the discussion.

Minor comments are listed below:

1)      When you talk about naming I think you should used labelling (eg: line 132)

2)      The correspondence between Fig 1 and Fig 2 is not clear with the description between lines135-139. I encourage you to draw an schema showing the time interval correspondences you have defined to group and determine rules.

3)      Why did you chose patients with at least one follow-up visit? Why not choosing at least 3 visits or a relative value (Q1-Q3)?

4)      Table 1 is confusing as it is now. It is number of patients or number of observations? For process mining and conformance it is of utmost important to know the number of observations and their distribution. I presume a chart will be easier to transmit the distribution of both number of patients and events for each AJCC stage.

5)      Table 2 shows a skewed distribution of fitness and precision. Average values are not good descriptors of skewed distributions, please include median and kurtosis.

6)      There are previous studies trying to tackle the problem of melanoma surveillance with machine learning methods: doi: 10.1109/ICCH.2012.6724482 ; doi: 10.1016/j.jaad.2017.09.055. Authors are encouraged to build on top of these initiatives and explain why conformance checking is needed.

7)      Please include ref 9 in line 63.

8)      In related work authors state that “precision metrics do not provide the desired properties to reliably recognize under-fitting” (line 82). How do you overcome this limitation?

9)      Model complexity should be also referenced, as for instance, doi: 10.1109/EMBC.2015.7318809

Author Response

The authors would like to thank the editor and the reviewer for their precious time and comments. We have carefully addressed all the comments. The corresponding changes and refinements made in the revised paper are summarized in our response below.

Comments and Suggestions for Authors

Rinner et al. Present a study of conformance checking in the follow-up of Melanoma using process mining techniques. The paper is well written and it is interesting. I enjoyed reading it and I am quite confident it can help epidemiologists, statisticians, doctors and engineers in producing advanced decision support systems.

I have only one major concern and it is about the formal structure as a scientific paper. The main shortcoming I have detected during the review is the thread between methods and results. On its current form, the manuscript presents and describes some results which were not presented in the objective nor the methodology, leading to a potential confusion and lack of reproducibility. Authors should complete the methods section to describe:

· Precision and fitness measurements

→ We moved parts of section 4.1 to the methods section to improve the thread between results and methods.  

· penalties (weights) configuration (line 153)

→ We described the penalties and the configuration in the methods section.

· define what a trace is and what an event is

→ We added the definition of traces and events to section 1.2 where it is first mentioned.

· Define how you aim to analyse survival

→ We added a section Statistical Methods and introduce the survival analysis there.

· Define why, how and for what did you used DISCO instead of ProM.

→ We added a paragraph about Disco and the task we use it for (mainly explorative analysis).

· Finally, discussion section should contain the authors viewpoint of the results and comparison with the related work and state of the art.

→ We think compliance calculated and formalized using the fitness using MPE is very promising. The our work is set into relation to the related work and state of the art as part of the paper.

· Including in the discussion new information (Fig 8) is misleading. Please move this chart and its description to results and only interpret it in the discussion.

→ We moved the Medical implications to the results section.

Minor comments are listed below:

1) When you talk about naming I think you should used labelling (eg: line 132)

→ We are now talking about labelling and not naming since “it” is attached to something and not indicating a thing.

2) The correspondence between Fig 1 and Fig 2 is not clear with the description between lines135-139. I encourage you to draw a schema showing the time interval correspondences you have defined to group and determine rules.

→ We changed the nomenclature in Figure 1 to match the nomenclature of Figure 2 to clarify the connection between the two figures. The time interval was used during the presentation of the paper yet we think it does not add sufficient value to warrant an additional figure.

3) Why did you chose patients with at least one follow-up visit? Why not choosing at least 3 visits or a relative value (Q1-Q3)?

→ In the Results section we added a sentence why patients without any follow-up visit were not included. “since patients without a single follow-up visit only had the excision at the DDMUV and no data is available in the melanoma registry”.

4) Table 1 is confusing as it is now. It is number of patients or number of observations? For process mining and conformance it is of utmost important to know the number of observations and their distribution. I presume a chart will be easier to transmit the distribution of both number of patients and events for each AJCC stage.

→ As indicated in the caption it is the number of patients. We added the mean number of events per case to the table 1 to get a feeling of the number of observations.

5) Table 2 shows a skewed distribution of fitness and precision. Average values are not good descriptors of skewed distributions, please include median and kurtosis.

→ To clarify this, we removed “Avg.” from precision since it is not really an average but a calculated metric (#Observed/#Possible Behaviour). Histograms and medians were not added since we did not see the additional value.

6) There are previous studies trying to tackle the problem of melanoma surveillance with machine learning methods:

doi: 10.1109/ICCH.2012.6724482 (An examination of TNM staging of melanoma by a machine learning algorithm) ;

doi: 10.1016/j.jaad.2017.09.055. (Machine learning and melanoma: The future of screening) Authors are encouraged to build on top of these initiatives and explain why conformance checking is needed.

→ We added a paragraph to the discussion, discussing how neural networks are currently used in the diagnosis of melanomas and that the triage of patients using neural networks could support physicians in their guideline compliant treatment.

7) Please include ref 9 in line 63.

→ Reference inserted.

8) In related work authors state that “precision metrics do not provide the desired properties to reliably recognize under-fitting” (line 82). How do you overcome this limitation?

→ We clarified the paragraph in the related work section, added the precision calculation paper to the literature and related to it in the methods section 3.2.

9) Model complexity should be also referenced, as for instance, doi: 10.1109/EMBC.2015.7318809 (Diabetes care related process modelling using Process Mining techniques. Lessons learned in the application of Interactive Pattern Recognition: coping with the Spaghetti Effect)

→ In this work the event log is generated based on event data in conjunction with a predefined model. A simplified version of this model is depicted in Figure 2. The Spaghetti Effect is a result of highly unstructured processes. This does not apply in our case and we think that model complexity in general is not an issue for this specific paper.